# A Survey of Awareness of Parents and Caretakers on Diagnostic Radiological Examination Related Radiation Exposure in a Tertiary Hospital in Malaysia

**DOI:** 10.3390/ijerph19073898

**Published:** 2022-03-25

**Authors:** Chee Guan Ng, Hanani Abdul Manan, Faizah Mohd Zaki, Rozman Zakaria

**Affiliations:** 1Department of Radiology, Faculty of Medicine, Hospital Canselor Tuanku Muhriz, Universiti Kebangsaan Malaysia Medical Centre, Kuala Lumpur 56000, Malaysia; henryncg@gmail.com (C.G.N.); drfaizah@ppukm.ukm.edu.my (F.M.Z.); rozman@ppukm.ukm.edu.my (R.Z.); 2Makmal Pemprosesan Imej Kefungsian, Department of Radiology, Faculty of Medicine, Pusat Perubatan Universiti Kebangsaan Malaysia, Kuala Lumpur 56000, Malaysia; 3Department of Radiology and Intervency, Hospital Pakar Kanak-Kanak (Children Specialist Hospital), Universiti Kebangsaan Malaysia, Kuala Lumpur 56000, Malaysia

**Keywords:** parents, caretakers, paediatric, radiation awareness

## Abstract

Background: With the advancement in medical imaging, radiological application in the paediatric population has also increased. Children, generally more radiosensitive, have a higher risk of developing certain malignancies. Therefore, this may result in uneasiness among parents and caretakers when their children need to undergo medical imaging examination. Hence, this study aims to assess the awareness of parents’ and caretakers’ awareness of medical imaging-related radiation exposure in our institution and their opinion of a medical radiation exposure-tracking programme for the paediatric population. Methodology: A cross-sectional survey was conducted for 6 months duration among parents and caretakers, who brought their children (under 12 years old) for imaging. The questionnaire booklet had eleven knowledge-based questions to assess respondents on ionising radiation-associated medical imaging, the radiation-related risk and radiation safety precaution. Results: Two hundred and fifteen respondents participated in this survey. More than 40% of the respondents failed to identify various dose-saving and ionising radiation-related imaging methods. Only 87 participants (40.5%) could correctly answer at least six out of eleven knowledge-based questions. Moreover, 88.4% of the respondents support a medical radiation exposure-tracking programme for their children. Conclusion: Parents and caretakers who visited our institution had inadequate awareness of medical radiation exposure. Appropriate measures need to be taken to address this promptly. Implementation of a medical radiation exposure-tracking programme for the paediatric population is considered timely as most respondents agree with this programme.

## 1. Introduction

Globally, more than 3.6 million diagnostic radiological examinations were performed annually from 1997 to 2007 [1,2]. Similarly, in Malaysia, all medical examinations also showed an increasing trend from 1990 to 1994 [3,4]. With the rapid advancement in imaging technology, the utilisation of these imaging modalities in the paediatric population has also increased [1,3,4]. UNSCEAR 2013 report showed that approximately 3 to 10% of the medical diagnostic radiological procedures were performed on children [1,5]. Cross-sectional imaging, for example, computed tomography (CT scan) has helped diagnose and manage care for the paediatric population [6].

Some radiological examinations utilise ionising radiation such as radiography, fluoroscopy, CT and radionuclide study [7], radiation is often associated with its potential biological effect from cellular damage [7]. Children who have a longer life span and are more radiosensitive have a greater risk of developing certain cancer due to radiation exposure [1,8,9]. Hence, the word “radiation” might cause uneasiness among the public [10,11,12]. Events such as the Chernobyl accident further aggravate fear among them [11,13].

One study by Ria F et al. [14,15,16,17] reported that patients had little awareness concerning radiation imaging modalities. Apart from patients, few studies also revealed poor awareness among healthcare professionals. This includes the medical radiation exposure and the associated risk and radiation protection to both patients and healthcare professional [15,16,17,18]. Healthcare professionals and the public might perceive differently on radiological examinations related to radiation risk [1,12]. Patients’ understanding of radiation would influence their agreement on diagnostic imaging procedures [12,18,19]. Effective communication between healthcare professionals and patients is the key to improving the experience and facilitating decision making [1,10]. Studies have shown that patients do not receive adequate information and do not have sufficient discussion regarding the benefits and risks of radiation in medical imaging [20,21,22]. Apart from doctors, radiographers also do not provide adequate information regarding medical imaging-related radiation doses and risks to patients [23]. Besides that, there is a paradigm shift of the radiation protection concept to individual dose and radiation exposure tracking to prevent excessive radiation dose [24]. The International Atomic Energy Agency (IAEA) introduced the Smart Card project in 2006 for patients’ radiation exposure tracking [25]. Tracking a patient’s radiation exposure will benefit patients, clinicians and radiologists as this could ensure patients receive minimal radiation but at the same time do not compromise optimal care [26].

Few studies have highlighted parents’ and guardians’ poor awareness of medical radiation risk [27,28]. However, to our best knowledge, no local research in Malaysia has been conducted to assess the awareness of parents and caretakers on diagnostic radiological examinations related to radiation exposure. Therefore, the present study aims to determine the awareness among parents and caretakers who visited our institution on medical imaging-related radiation exposure. This study includes a survey on their opinion of having to be involved in a medical radiation exposure-tracking programme for their children. It will be the initial starting point to evaluate whether the radiation exposure-tracking program is necessary for the future.

## 2. Materials and Methods

### 2.1. Study Design

A cross-sectional survey was conducted among parents and caretakers using self-administered questionnaires in a tertiary teaching hospital. This survey followed the study protocol approved by a local institutional research and ethics committee. All parents and caretakers who brought their children to undergo radiological examination in the Radiology Department were invited to participate in this study. Authors defined paediatric patients as children from newborn to 12 years old. Any parents and caretakers who refused to consent or with incomplete questionnaires were excluded from this study.

### 2.2. Questionnaire Design

This questionnaire booklet was designed with a joint effort of the authors, a paediatric radiologist, a physicist and a biostatistician. It was set in the national language, Bahasa Melayu and made up of questions requiring a simple tick and short answer.

This questionnaire booklet had eleven knowledge-based questions. These questions were deemed statistically reliable and validated for the objective after a pilot study was performed among sixty-five parents and caretakers. The reliability analysis assessment using internal consistency was acceptable with Cronbach’s alpha value of 0.700 (95% CI: 0.579–0.798). These questions were also analysed using item difficulty index, item discrimination index and exploratory factor analysis.

The questionnaire booklet was divided into two sections. In Section A, questions encompassed demographic information such as gender, age, ethnic group, occupation and education level. In Section B, there were eleven knowledge-based questions. Seven questions required the respondents to identify dose-saving and ionising radiation-related medical imaging (Questionnaire booklet: Section B, Question no. 3). The other four questions assessed respondents’ perception of radiation in medical imaging, radiation-induced cancer risk and radiation safety precaution (Questionnaire booklet: Section B, Questions no. 4–7). This study divides diagnostic radiological imaging into dose-saving and ionising radiation-related medical imaging. Dose-saving imaging modalities refer to those that do not utilise ionising radiation, namely ultrasound and MRI.

Besides that, in Section B, there were questions to survey if respondents received information regarding medical imaging that their children underwent from the respective doctors. In the same section, we also asked their opinion of having to participate in a radiation exposure-tracking programme for the paediatric population.

### 2.3. Data Collection

This study was conducted six months from 1 June 2020 to 31 December 2020. Hardcopy questionnaires were distributed to parents and caretakers who visited the Radiology Department. Informed written consents were obtained from the respondents, and they were required to complete the questionnaire in one setting without any references.

### 2.4. Statistical Analysis

The data were analysed using IBM SPSS Version 22. Descriptive analyses such as frequency and percentage were used to analyse demographic data and to determine the level of knowledge on ionising radiation-related imaging modalities and perception of cancer risk.

To assess the overall level of knowledge, an arbitrary one point was given to each correctly answered knowledge-based question. Then, further analyses were conducted using chi-square and Spearman correlation to determine the association between knowledge level and different demographic variables such as respondents’ roles. The statistically significant level was set as a *p*-value < 0.05.

## 3. Results

Three hundred hardcopy questionnaires were distributed during the six months. However, only 215 respondents had completed and returned the questionnaire booklet, giving a response rate of 71.7%.

Out of two hundred and fifteen respondents, 139 females (64.9%) and 76 males (35.3%). Most of them were from the Malay ethnic group (76.7%). Mothers made up more than half of the total respondents (*n* = 117 persons), and 74.5% were from the 30 to 49 age group. 83 (38.6%) of the respondents had received tertiary education such as degree, master’s degree and Ph.D. Respondents’ occupations also varied, including clerks, healthcare workers, homemakers, and educators (Table 1). More than 70% of the respondents (*n* = 154) reported that they had heard of radiation before this survey.

On the knowledge of identifying ionising radiation-related imaging modalities, this study found that less than 50% of the respondents knew that fluoroscopy, CT and radionuclide study are associated with ionising radiation (Table 2). The results also show that only 97 respondents realised that CT uses a higher radiation dose than radiography. Concerning ultrasound and MRI, 34.9% (*n* = 75) and 15.8% (*n* = 34) of the respondents were aware that these modalities are dose saving, respectively (Table 3). However, 70.7% (*n* = 152) of them agreed that ultrasound is generally a safe imaging modality for children.

On the perception of cancer risk associated with ionising radiation, only 41.9% of the total respondents (*n* = 90) agreed that exposure to ionising radiation could increase the risk of developing cancer. For radiation safety, more than 70% (*n* = 165) of our respondents understood that pregnant women are not allowed to accompany their children during chest radiography examinations.

On overall knowledge, the median score for the 11-item knowledge-based questions was 4.6. Authors arbitrarily set the satisfactory knowledge level as respondents who could answer at least six knowledge-based questions (out of 11). There were 87 participants (40.5%) who achieved acceptable knowledge scores, and a significant association was demonstrated between the respondents’ role and the level of knowledge (Pearson chi-square = 8.952, *p*-value = 0.011) (Table 4). There was no significant association between the knowledge score and other demographic variables such as age group, ethnicity, and gender.

One hundred and eighty of the respondents (83.7%) agreed that their doctors would explain their children’s radiological examination. However, there is no verifiable significance between the explanation frequency and knowledge level (Pearson chi-square = 4.215, *p*-value = 0.239) (Table 5). Related to the medical radiation exposure tracking programme, 88.4% of the respondents (*n* = 190) strongly agreed and agreed to have the tracking programme, which allows them to trace the medical radiation exposure and type of radiological examination their children underwent.

## 4. Discussion

The present study is the first study in Malaysia that has been conducted to survey the awareness on radiation knowledge and radiation safety of caretakers during diagnostic radiological examination in a tertiary hospital. This study also aims to explore the opinion of having a medical radiation exposure-tracking programme for children. There was no published report related to knowledge on radiation and radiation safety of Malaysian parents and caregivers. Therefore, this study is significant to be carried as for the baseline data for further research in the future. Most importantly, with the present finding, healthcare professionals need to work harder to educate the public, especially parents and caretakers of young populations.

Children are more radiosensitive and have a higher risk of developing malignancy due to radiation exposure [1,8,9]. Knowing this fact, parents and caretakers need to have basic knowledge about radiation, such as the type of imaging their children undergo. A previous review by Ribeiro et al. [20]. Showed that patients generally lack knowledge on radiation exposure and safety The present study found that only 40.5% of the respondents achieved a satisfactory knowledge score on medical imaging-related radiation exposure and safety. More than 40% of the respondents could not identify various dose-saving and ionising radiation-related imaging modalities. It is worth noting that 41.4% (*n* = 89) were unaware that radiography is associated with radiation. Although the radiation dose for radiography is low, the considerable collective risk cannot be ignored as the total number of radiographic examinations is high [29,30].

Interestingly, caretakers showed better awareness of medical radiation than parents with a significant association demonstrated. The postulated explanation is that caretakers, including grandparents and siblings (15.8%), might have prior experience on this radiological imaging, and siblings might have learned about this information from school. Besides that, respondents who work in the healthcare-related sector generally better understand this subject matter. There were thirty healthcare worker respondents in this survey, and twenty-five achieved satisfactory knowledge scores. This might be due to their experience related to medical radiation during work.

More than 70% of the respondents understood that pregnant women are not allowed to accompany their children during the radiography procedure. The presence of warning signboards displayed on each X-ray and CT room in this centre helps remind the parents and caretakers. In addition, radiographers in our centre would further check on the last menstrual period of all female patients and companions before conducting radiation-associated imaging. This indirectly improves parents’ and caretakers’ awareness of this subject matter.

Many studies highlighted a lack of communication between healthcare professionals and patients regarding medical radiation exposure and the associated risk [20,21,22]. The present study found that 83.7% of the respondents agreed that their doctors explained various frequencies on the medical imaging procedures their children would undergo. However, parents and caretakers who received this information did not necessarily have satisfactory awareness of medical imaging-related radiation exposure and safety. Possible reasons include simplified information and time constraints in the clinic, further limiting healthcare professionals from detailed explanations [22,27].

A majority (88.4%) of the parents and caretakers agreed to a radiation exposure-tracking programme. This will allow them to trace radiation exposure and the type of radiological examination that their children underwent. In addition, more than 85% (*n* = 193) of the respondents were willing to bring a booklet or card for recording purposes whenever their children undergo a radiological medical examination. With the support from parents and caretakers, a radiation exposure-tracking programme could be conducted in the future. Children are at a greater risk than adults to develop cancer after being exposed to radiation and have the probability of living longer than the adult population [28,29,30]. Therefore, it is crucial to have a medical radiation exposure-tracking programme. Furthermore, some of the children will go to different hospitals from time to time. With the implementation of this system, we would be able to trace the radiation dose for the particular paediatric patient.

There were a few limitations in the current study. The application of convenience sampling in this study may result in sampling bias. Due to the COVID-19 pandemic, there was also difficulty recruiting respondents to participate in this survey. Besides that, as the study was conducted in a single centre, the result was not representative of the whole country. A multi-centre survey is recommended for future research to generate a more representative result.

In conclusion, parents and caretakers in our institution have inadequate awareness of medical imaging-related radiation exposure. Besides that, the presence of communication but ineffective may contribute to parents’ and caretakers’ poor understanding of medical imaging-related radiation exposure. We would like to recommend that doctors and radiographers spend more time explaining to parents and caretakers related to this radiation safety and risk. Apart from the explanation given by healthcare providers, the educational brochure could be distributed to them as reading materials while waiting for imaging in the radiology department. The ultimate way to effectively track medical radiation exposure, radiation dose, and type of paediatric imaging patients underwent is to implement a radiation-tracking programme. This study has shown that our respondents were supportive of this programme.

## 5. Conclusions 

Some of the *Take Home Points* includes more than 40% of the parent and caretaker respondents failed to identify dose-saving and ionising radiation-related imaging modalities. Only 40.5% of the respondents achieved a satisfactory knowledge score on ionising radiation associated medical imaging, its related risk and radiation safety. There is a significant association between the respondents’ role and the level of knowledge. More than 80% of the parent and caretaker respondents received explanations on the medical imaging that their children would undergo. However, this does not improve their knowledge of medical imaging-related radiation exposure. Most of the respondents agreed to a radiation exposure-tracking programme for their children. Parents and guardians need to know the medical imaging their children are subjected to and the associated risk, particularly those related to ionising radiation.

## Figures and Tables

**Table 1 ijerph-19-03898-t001:** Demographics of respondents in the survey.

Demographic Variables	Number of Respondents/in Percentage, %, with Total Respondents: 215
Gender	
Male	76 (35.3)
Female	139 (64.7)
Relationship to the paediatric patient	
Father	64 (29.8)
Mother	117 (54.4)
Caretaker	34 (15.8)
Age Group	
20–29	31 (14.4)
30–39	110 (51.2)
40–49	50 (23.3)
50–59	11 (5.1)
60–69	12 (5.6)
70 and above	1 (0.5)
Ethnicity	
Malay	165 (76.7)
Chinese	32 (14.9)
Indian	9 (4.2)
Others	9 (4.2)
Education	
Primary education	4 (1.9)
Secondary education	64 (29.8)
Form 6/A level/diploma	60 (27.9)
Degree/master’s degree/Ph.D.	83 (38.6)
Others: not specified	4(1.9)
Occupations	
Clerk	19 (8.8)
Healthcare worker	30 (14.0)
Retiree	5 (2.3)
Housewife	40 (18.6)
Educator	12 (5.6)
Others	109 (50.7)

**Table 2 ijerph-19-03898-t002:** Questionnaire responses on ionising radiation-related medical imaging. It can be seen that 58.6% of the respondents (*n* = 126) know that radiography is associated with ionising radiation. For other medical imaging such as fluoroscopy, CT, radionuclide study and angiography, more than 50% of the respondents did not realise these modalities involve the utilisation of ionising radiation.

Questions	Number of Respondents
1. Radiography uses ionising radiation?	
Yes	126 (58.6%)
No/Unsure	89 (41.4%)
2. Fluoroscopy uses ionising radiation?	
Yes	41 (19.1%)
No/Unsure	174 (80.9%)
3. Computed tomography uses ionising radiation?	
Yes	89 (41.4%)
No/Unsure	126 (58.6%)
4. Radionuclide study uses ionising radiation?	
Yes	92 (42.8%)
No/Unsure	123 (57.2%)
5. Angiography (fluoroscopy) uses ionising radiation?	
Yes	50 (23.3%)
No/Unsure	165 (76.7%)

**Table 3 ijerph-19-03898-t003:** Questionnaire responses on dose-saving medical imaging, namely ultrasound and MRI. More than 50% of the respondents did not realise these modalities are not associated with ionising radiation.

Questions	Number of Respondents
1. Ultrasound uses ionising radiation?	
No	75 (34.9%)
Yes/Unsure	140 (65.1%)
2. MRI uses ionising radiation?	
No	34 (15.8%)
Yes/Unsure	181 (84.2%)

**Table 4 ijerph-19-03898-t004:** The number of respondents who achieved satisfactory knowledge score and their relationship with the paediatric patients. There is a significant association between respondents’ roles and knowledge on the subject matter.

Relationship to Paediatric Patients	Level of Knowledge (out of 11 Knowledge-Based Questions)		Total
	<6	≥6	
Father	37	27	64
Mother	78	39	117
Caretakers	13	21	34
Total	128	87	215

(Pearson chi-square = 8.952, *p* value = 0.011).

**Table 5 ijerph-19-03898-t005:** The number of respondents who achieved satisfactory scores and the frequency of explanation regarding radiological examinations that their children underwent. Ninety-nine respondents agree that doctors would explain every medical imaging ordered for their children. There is no association between frequency of explanation and level of knowledge level on the subject matter.

Frequency of Explanation by Doctors on Radiological Examination	Level of Knowledge (out of 11 Knowledge-Based Questions)		Total
	<6	≥6	
Never	26	9	35
Sometimes	30	24	54
Most of the times	14	13	27
Explain every ordered imaging.	58	41	99
Total	128	87	215

(Pearson chi-square = 4.215, *p* value = 0.239).

## Data Availability

The authors declare that they had full access to all of the data in this study and the authors take complete responsibility for the integrity of the data and the accuracy of the data analysis.

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
