# Peer review of "A Survey of Awareness of Parents and Caretakers on Diagnostic Radiological Examination Related Radiation Exposure in a Tertiary Hospital in Malaysia"

_ijerph, 2022, doi:10.3390/ijerph19073898_

Round 1
Reviewer 1 Report
Reviewing a paper entitled: A Survey Of Awareness Of Parents And Caretakers On Diagnostic Radiological Examination Related Radiation Exposure In A Tertiary Hospital In Malaysia
Comments:
The paper is well written and structured.
I have very few comments:
According to the questionnaire related to the modalities associated with ionising radiation listed in Table 2, angiography is not a modality, but a function that can involve various modalities, including modalities without ionising radiation such as MRI. This is somewhat confusing, so I would suggest to use fluoroscopy.
Author Response
|
No. |
Comments/Suggestions
|
Response |
|
|
Reviewer #1:
The paper is well written and structured.
I have very few comments:
According to the questionnaire related to the modalities associated with ionising radiation listed in Table 2, angiography is not a modality, but a function that can involve various modalities, including modalities without ionising radiation such as MRI.
This is somewhat confusing, so I would suggest to use fluoroscopy.
|
We have done the corrections related to this angiography. The modifications have been highlighted in blue from page 8.
Fluoroscopy has been used in the whole results and discussion as suggested. Amendment also has been done in Table 2. |

Reviewer 2 Report
It is important to investigate the awareness of parents and caretakers for medical radiation exposure. The study has clarified the knowledge of radiation exposure for parents using a questionnaire. However, there are some issues in this manuscript. Please check below for details.
Major comment
- Your considerations are often a repetition of the results. Compare with the references and add novelty to your paper.
- What exactly is the dose-tracking program that you are advocating? Is it going on in your country specifically? If so, please describe the trend.
Minor comments
- You should write the capitalization rules.
- The black circles in the table are unnecessary.
- p 6, line193-195. Please indicate the specific percentage of siblings and grandparents.
- The references do not follow the instructions for authors at all. Please carefully revise them.
Author Response
|
|
Reviewer #2:
It is important to investigate the awareness of parents and caretakers for medical radiation exposure. The study has clarified the knowledge of radiation exposure for parents using a questionnaire. However, there are some issues in this manuscript. Please check below for details.
Major comment
Your considerations are often a repetition of the results. Compare with the references and add novelty to your paper.
What exactly is the dose-tracking program that you are advocating? Is it going on in your country specifically? If so, please describe the trend.
Minor comments
1. You should write the capitalization rules.
2. The black circles in the table are unnecessary.
3. p 6, line193-195. Please indicate the specific percentage of siblings and grandparents.
4. The references do not follow the instructions for authors at all. Please carefully revise them.
|
The importance of the study needing to be carried out has been added to page 9. The added information is in blue.
The information related to this dose-tracking program has been added according to reviewer suggestions on page 11. The added information is in blue.
1. We are very sorry, we do not understand the suggestions. We would like to have a further explanation of these particular comments.
2. The black circle in the table has been removed accordingly.
3. The specific percentage of siblings and grandparents has been added on page 10. The added information is in blue.
4. We have revised the reference accordingly as per reviewer suggestions. |

Round 2
Reviewer 2 Report
Thank you for revising the report.
The references do not follow the instructions for authors at all. Please carefully revise them.
Author Response
Dear reviewer,
Thank you for the comment and suggestion. Attached we have done the correction for the reference accordingly. The corrections are in red.
Best regards,
Hanani
